# Review on the Art of Roof Contacting in Cemented Waste Backfill Technology in a Metal Mine

Fengbin Chen [1,2], Jiguang Liu [1,2], Xiaowei Zhang [3], Jinxing Wang [1,2,*], Huazhe Jiao [1,2,*] and Jianxin Yu [1,2]

1   School of Civil Engineering, Henan Polytechnic University, Jiaozuo 454003, China; fbchen@hpu.edu.cn (F.C.); liujiguang1998@126.com (J.L.); jianxinyu@hpu.edu.cn (J.Y.)
2   Collaborative Innovation Center of Coal Work Safety and Clean High Efficiency Utilization, Jiaozuo 454000, China
3   Jiaozuo Qianye New Material Co., Ltd., Jiaozuo 454150, China; zhangxiaowei@163.com
*   Correspondence: wjx@hpu.edu.cn (J.W.); jiaohuazhe@hpu.edu.cn (H.J.)

**Abstract:** The backfilling mining method can effectively solve the environmental and safety problems caused by mining. It is the key technology to realize green mining. Scientific development has accelerated the pace of research on the rational utilization of mine solid waste and improved the research level of backfilling technology. The development history of the backfilling mining method is introduced in the present paper, and it is determined that roof-contacting backfilling is the key technology of mine-solid-waste backfilling mining. This paper introduces three calculation methods of similar roof-contacted backfilling rates. In this paper, the relationship between the characteristics of backfilling slurry made from solid waste from mines and the roof-contacted backfilling rate is systematically analyzed, such as the flow law in stope (gravity gradient), bleeding shrinkage, and natural sedimentation of backfilling slurry. It is pointed out that the characteristics of the stope, such as washing-pipe water, water for the leading way, filling pipeline, and shape of the backfilling stope, are closely related to the roof-contacted backfilling rate. The influential relationship between objective factors, such as human factors, limited auxiliary leveling measures, and backfilling "one-time operation" in the backfilling process, and high-efficiency top filling are considered, and a schematic diagram of the influencing top-filling rate and structure is drawn. At the same time, this paper summarizes the improvement measures of roof connection from three aspects. These include the use of expansive non-shrinkable materials, forced roof-contacted technology, and strengthening management level. It is pointed out that the roof-contacted filling technology is still facing severe challenges, and the research on the backfilling connection technology needs to be strengthened.

**Keywords:** solid-waste filling; roof-contacted rate; influencing factors of roof connection; regulation and improvement of roof-connection measures



## 1. Introduction

Mineral resources are the precious materials given to mankind by nature and the basis for the survival and development of human society. They are closely and positively proportional to the enhancement of national economic strength, civilization and progress, social stability, and the improvement of national living standards [1–3]. Whether from the Stone Age or the Information Age, a kind of mineral raw material with good performance and strong functionality is produced in each historical development stage. Therefore, without the development and utilization of mineral resources, human society will not be able to progress [4–7]. However, the safety and environmental problems accompanying the mining process restrict the sustainable development of mineral resources [8–10].

China's economic take-off is inseparable from the great help of mining. Due to the combination of extensive type and excess capacity production, the area of land destroyed by mining in China has reached 2 million $hm^3$. The amount of tailings formed by mining and

beneficiation has reached 14.6 billion tons, occupying a land area of 8700 km$^2$, equivalent to the area of four Shenzhen cities [11]. About 5–7 billion tons of tailings are generated globally every year [12–14]. The generation of such a large quantity of mill tailings adversely affects the environment, including the air, water, and soil [15–18], as shown in Figure 1. At the same time, in the process of underground mining, a large number of mined-out areas are created, which not only threatens the safety of underground operations, but also includes the hidden danger of inducing mine earthquakes and surface collapse [19–21]. To solve a series of problems caused by mining, such as surface stacking, solid waste stacking, and goaf treatment, it is necessary to break through the bottleneck constraints of resources and the environment, adhere to energy conservation and emission reduction, and develop a mining circular economy [22,23]. With the idea of "one filling to treat three wastes, one waste to treat two hazards", the filling technology creatively uses mine solid waste efficiently, eliminates the tailings pond, and governs the goaf, forming a mining method with a high recovery rate and low dilution rate [24–26]. It is an important technical means to move towards the win–win situation of building "environmentally friendly" and "safe and efficient" mines [18,27–30]. Nowadays, there are many studies on the strength of backfilling massif, but there are relatively few studies on how to improve the roof-contacted backfilling rate to address the overall performance of the backfilling massif, which is an urgent problem that needs to be solved.

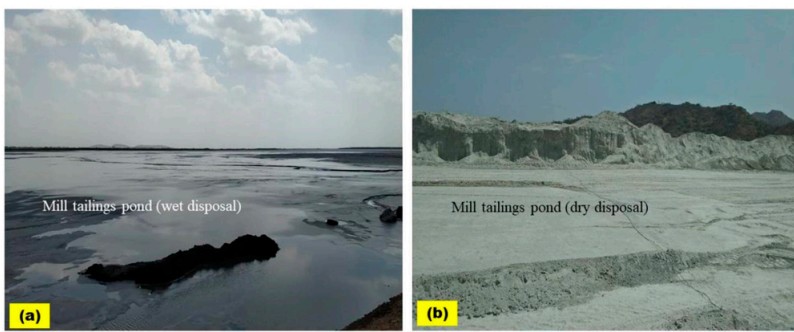

**Figure 1.** Surface disposal of tailings: (**a**) Wet disposal and (**b**) dry disposal.

## 2. Development of Backfilling Technology and Roof-Contacting Backfilling Technology

The backfilling mining method has a development history of more than half a century. At first, it aimed at the simple treatment of mine solid waste, such as waste rock. Nowadays, backfilling mining has gradually developed into a comprehensive technology to improve the environment, control the ground pressure, and reduce the poor-loss index to form a complete mining process [31,32]. According to different backfilling materials, backfilling mining has experienced dry backfilling, hydraulic sand backfilling, and cemented backfilling [33–38]. From dry backfilling to cemented backfilling, backfilling technology has developed rapidly and gradually developed to include paste filling, as shown in Figure 2.

Due to the different mining methods, in some upward-filling mining methods, the roof-contacted backfilling rate is not strictly required, such as the upward-approach filling mining method. However, in the continuous mining and continuous backfilling method and open stoping following the backfilling mining method, the roof-contacted backfilling rate is of great concern, which is directly related to the ability of the upper surrounding rock to be a form of support, and thus ensure the safety of the stope [39–41]. Over the years, the relevant experts believe that, for the conventional high-concentration cemented filling, the influences of roof-contacting backfilling are mainly gravity gradient, bleeding shrinkage, and natural sedimentation of backfilling slurry. [42–44]. In order to achieve an efficient one-time roof connection, a series of processes, such as setting multiple backfilling discharge openings and the shape of stope backfilling adjustment, are proposed, but the practical application effect is poor.

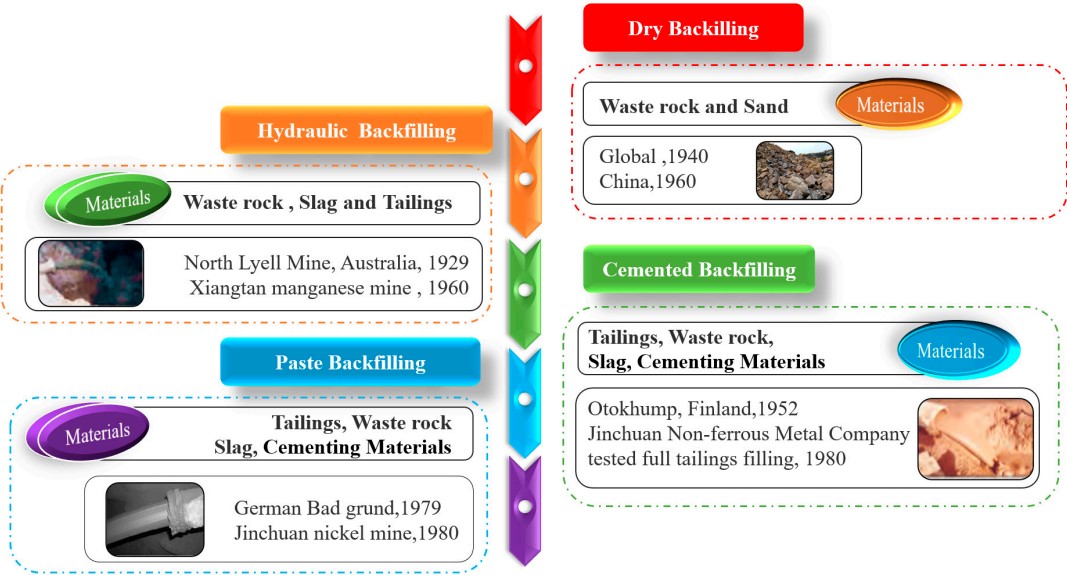

**Figure 2.** Development history of backfilling mining method.

Therefore, in order to improve the roof-contacted backfilling rate, the safety of the mining operation, and reduce the environmental burden, this paper analyzes the factors affecting the roof-contacted backfilling rate under the conventional high-concentration cemented filling, and summarizes the corresponding improvement measures.

## 3. Factors Affecting Roof-Contacted Backfilling Rate and Improvement Measures

### 3.1. Similar Roof-Contacted Rate

The roof contact of the backfilling massif is an important part of backfilling work. The effect of the roof connection of the backfilling massif can be expressed by the backfilling roof-contacted ratio. This definition can be intuitively understood as the ratio of the area of the filling body contacting the roof to the area of the whole roof. However, in practical applications, this method is difficult. In order to simplify the calculation, three kinds of similar methods are proposed in the literature to calculate the roof-contacted backfilling rate, as shown in Table 1 [45].

**Table 1.** Similar roof-contacted rate.

| Method | Calculation Formula | Explanation |
|---|---|---|
| Similar volume ratio | $\epsilon = \frac{v}{V} \times 100\%$ | $\varepsilon$—Similar backfilling roof-contacted rate<br>$v$—Volume of filling body<br>$V$—Volume of mined ore |
| Average height ratio | $\epsilon = \frac{h}{H} \times 100\%$ | $H$—Average height of measured area before filling<br>$h$—Average height of filling body in the measured area |
| Cross-sectional area ratio | $\epsilon = \frac{s}{S} \times 100\%$ | $s$—The area of the top of the filling body connected to the top<br>$S$—Area of goaf roof |

### 3.2. The Influencing Factors of Roof-Contacted Backfilling Rate

3.2.1. Slurry Characteristics

(1)    Gravity gradient

In the past, filling and mining experts have performed a large number of similar simulations and field industrial tests for the roof connection of the backfilling massif, conducted a lot of research on backfilling slurry accumulation contour and slope angle, and summarized and proposed a significant amount of application experience [46–48]. If self-flowing transportation is adopted, the backfilling slurry shrinkage can be divided

into two processes, namely, the unrestricted sediment diffusion process and the restricted upward-stacking process.

The unrestricted sediment diffusion process refers to the filling slurry that does not make contact with the boundary of the short side of the corresponding route after the filling slurry is filled into the goaf, and it can be regarded as the free flow of slurry on an infinite plane. It can be observed from Figure 3 that the curve law of free flow on the infinite plane has the characteristics of normal distribution. The sediment diffusion movement of aggregates with different particle sizes in the filling slurry can be regarded as sedimentation diffusion events, and the successive sedimentation along the goaf presents probabilities.

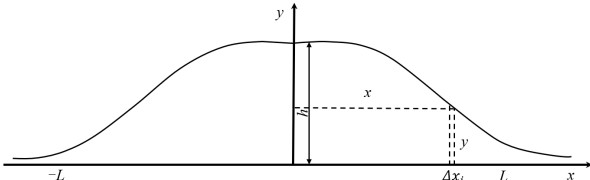

**Figure 3.** Sediment diffusion slope curve with unrestricted slurry.

The individual aggregate molecules of the backfilling slurry are not enough to affect the whole backfilling massif. $k$ ($k = 1, 2, \ldots, n$) is used to represent the aggregate with different sizes constituting the filling slurry. According to the Lyapunov central limit theorem, sedimentary events within $\Delta x_i$ are always independent of each other. No matter what distribution the random variable $k$ has, when $n$ in $\sum\limits_{k=1}^{n} x_k$ approaches infinity, it approximately conforms to the normal distribution $N(0,1)$. From this, it is calculated that the slope curve equation of slurry infinite diffusion sediment is

$$y = he^{-\frac{x^2}{2\sigma^2}} \tag{1}$$

where $h$ is the maximum height of unrestricted diffusion sediment. $\sigma^2$ represents the mean squared deviation.

The mean squared deviation reflects the steepness and slowness of the slurry sedimentation slope, which is jointly determined by the slurry concentration, particle-size distribution, and the content of cementitious materials. The specific value of each mine should be estimated by the histogram of the density function through experiments. The filling slurry diffuses from the initial falling point to the surroundings to form a sedimentary body with an approximate radius of $L$. The slope of the sedimentary body is approximately normally distributed. There is a certain functional relationship between the unrestricted diffusion radius $L$ of the backfilling slurry in the stope and the sedimentary height $h$; that is, $h = (L)$.

The restricted upward-stacking process is a process in which the filling slurry enters the goaf, diffuses to the relatively short-side boundary, and accumulates upward until it is relatively close to the roof of the backfilling stope. After the backfilling is completed, by observing the final shape of the backfilling massif, it can be found that the effect of the backfilling slurry roof-connection in an area centered on the filling pipe's orifice is remarkable, while a certain sedimentary slope is formed at the edge of the area far from the pipe orifice, which is related to the backfilling slurry concentration, and mobility and operation times.

On the basis of the three-dimensional scanning of the basic-settlement-layer contour, Wang Xinmin et al. [47] used the model to estimate the roof-contacted area and roof-contacted backfilling rate, which provided a basis for the next safety production of the mine. Lu Hongjian et al. [49] studied the flow-trajectory model of backfilling parts in the filling stope of Shirengou Iron Mine, detected the surface contour of the backfilling massif by using three-dimensional laser scanning technology, and analyzed the characteristics of the backfilling-slurry-flow sediment slope curve, as shown in Figure 4. Tang Li et al.

studied [50] the problem of filling and roof connection in the Jinchuan No. 2 mining area. By taking the rod ground sand-based cemented filling slurry as an example, using the theoretical model, parameters, such as the optimal size of the stope, the optimal position of the filling pipeline, and the optimal number of pipeline movements, were studied to ensure the roof-connection effect of the filling body.

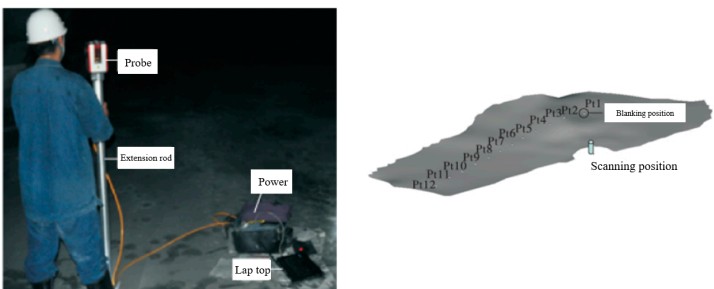

**Figure 4.** Flow path and slope monitoring of filling slurry in stope.

"The infinite sedimentary diffusion body model" assumes that the backfilling massif is the mean slurry without segregation, so it has good applicability when the stope size is small. However, in relatively large stopes, coarse aggregate of backfilling slurry sedimentate faster than fine aggregate, as shown in Figure 5 [51], so there are more coarse aggregates at the filling pipeline, and fine particles move and sedimentate to the far end with the movement of the slurry. The newly injected slurry flows and sedimentates along the slope of the deposited slurry. During the backfilling process, the gravity water accumulates at the far end of the underground stope and is discharged.

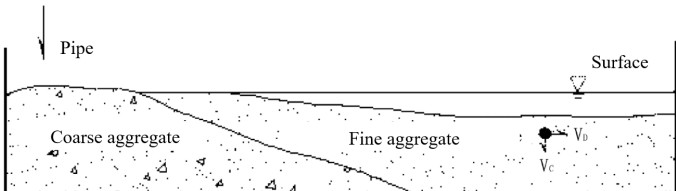

**Figure 5.** Sedimentation model of backfilling slurry with different aggregate sizes.

Hua-fu Qiu [52] modified the original model to some extent, and considered that the backfilling slurry expanded to both sides after entering the underground stope, and the mass fraction gradually decreased. The backfilling slurry still conforms to the normal distribution in the processes of flow and sedimentation. The modified slope curve model relationship is as follows:

$$y = a + he^{\frac{(x+b)^2}{2\delta^2}} \tag{2}$$

where $y$—slope height; $h$—sedimentation height of filling slurry; $\delta^2$—mean square deviation; and $a$, $b$—undetermined constant.

The undetermined constants $a$ and $b$ are related to the location of the backfilling area. When the backfilling slurry flows and sedimentates on the wireless plane, both $a$ and $b$ are 0. At this time, the slope curve model degenerates into the original infinite plane accumulation model.

(2)  Dewatering and sedimentation

Goaf filling and dewatering are an essential part of backfilling work transported by pipeline hydraulic gravity. A filling body is a kind of loose body, so its water content is complex, which can be divided into adsorbed water, capillary water, and gravity water [53]. Gravity water is the main object of dewatering in the backfilling process. This part of water exists in the large pores in the solid aggregate of backfilling. It has the general

characteristics of water and can flow freely between the backfilling aggregate and flow downward under the action of gravity [54].

Dewatering technology can be divided into two categories: external action and non-external action. The external effects mainly include the electro-osmosis method, negative pressure method, and pressure ventilation. The non-external effects mainly include the chain dewatering method, setting the dewatering closed wall and dewatering well, increasing the installation spacing and pipe diameter of a dewatering pipe, increasing the backfilling water overflow pipeline, and improving aggregate gradation [55].

Zhang Aiqing et al. [55] improved the common dewatering pipe without considering the external effect, increasing the number of dewatering pipes, and shortening the spacing of dewatering pipes, designed a new root-like dewatering pipe based on bionics, and conducted dewatering tests of the new root-like dewatering pipe and the common dewatering pipe, respectively. It was concluded that the new root-like dewatering pipe can significantly improve the dewatering rate compared tot the common dewatering pipe, as shown in Figure 6. Wang Bingwen et al. [56] explored the relevant laws of electro-osmotic dehydration and consolidation of filling slurry, and conducted the test with a self-made electro-osmotic dehydration test and natural dehydration test device, as shown in Figure 7. The results show that the electroosmosis method can not only accelerate the drainage speed, but also improve the strength of the test block for the full tailings non-cemented filling slurry without cementitious material.

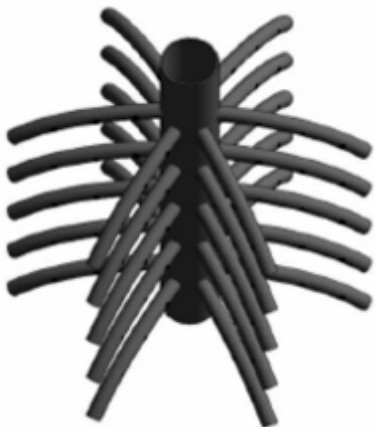

**Figure 6.** New type of dehydration pipe.

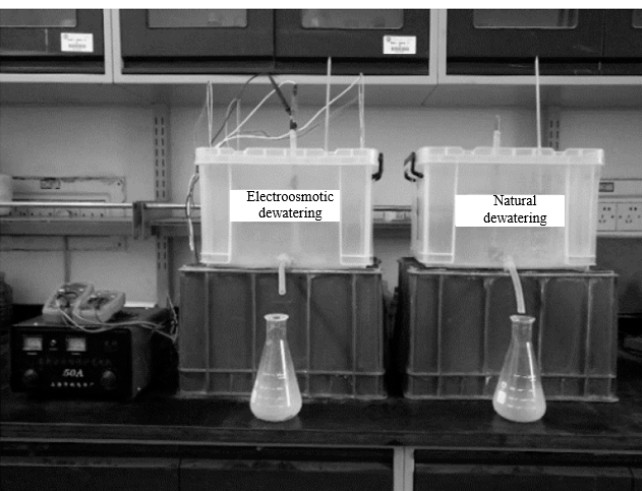

**Figure 7.** The test device.

(3)　Slurry shrinkage

The shrinkage of backfilling slurry is an inevitable phenomenon in the backfilling process, which is mainly in the form of pipeline hydraulic gravity and heterogeneous flow transportation. In order to ensure hydraulic gravity transportation, the water content of the slurry is much greater than that required by the hydration process of the cementitious material, and a large quantity of water needs to be removed in the filling process [57–59]. The shrinkage of backfilling slurry is mainly composed of bleeding shrinkage and seepage shrinkage.

Bleeding shrinkage is due to the fact that the backfilling slurry is in the state of supersaturated water. When the cemented filling slurry enters the goaf, the coarse and fine aggregates of the backfilling slurry sink one after another, forcing the rich water to separate on the surface of the backfilling massif. When the water is removed, a space is formed between the backfilling massif surface and the roof of the goaf, thus affecting the roof-contacted effect in filling process. Seepage shrinkage refers to the transition from a supersaturated state to a saturated or wet state after the water in the backfilling slurry is discharged by runoff. Following this, the gravity water in the gap between the solid aggregates is discharged by seepage, the aggregates are rearranged, the porosity between the solid aggregates is reduced, and the backfilling massif is subject to secondary sedimentation. In order to solve the bleeding problem of low-concentration self-flowing tailings filling slurry, Liu Juanhong et al. [60] conducted experiments with solid concentrations of 57%, 60%, 63%, and 66%, respectively. The results are shown in Figure 8., which verify that the bleeding and shrinkage of filling slurry causes the filling body to be unable to connect and compact.

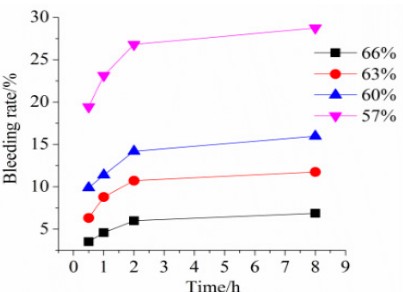

**Figure 8.** Effect of the different solid concentrations on slurry bleeding rate.

3.2.2. Stope Characteristics

(1)　Washing-pipe water and water for the leading way

In the hydraulic filling mode of gravity transportation, in order to prevent the residue of filling slurry left in the pipeline from sticking and ensure the smooth outflow of slurry, 5~10 min of water for the leading way and washing-pipe water are discharged before and after each filling. A large quantity of water cannot rapidly dewater from the stope in the stope. After the goaf is filled and dehydrated, a space is formed, which is difficult to ensure the roof connection of the backfilling massif.

(2)　Backfilling pipe

During the backfilling process of the goaf, the backfilling pipeline must be hung in the safety zone of the highest point of the goaf. Due to the limitation of stope conditions, it is difficult for the backfilling pipeline to be hung in the highest position. At this time, relevant measures, such as cutting, need to be taken to ensure that the backfilling pipe is safely set at the highest point of the stope. During the design, the position of the filling hole must be designed according to the mobility of the backfilling slurry. Certain measures, such as multiple backfilling, zone filling, reasonable setting of exhaust pipe and mobile backfilling pipe can be adopted to ensure the dense connection of the top. A mobile backfilling pipe is used in the Sanshandao gold mine to ensure that the filling massif is roof-contacted and

dense. Chen Qiusong et al. [61] used paste slurry as the research object, based on the actual stope size, and applied the similarity theory to design the simulated stope size, as shown in Figure 9. Based on the test results, they proposed reasonable suggestions for the position of the discharge port during staged backfilling to improve the backfilling roof-contacted effect.

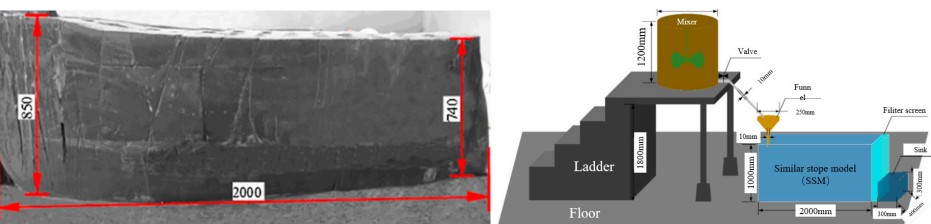

**Figure 9.** Simulated stope backfilling massif.

(3)  Shape of filling field

There are often no strict requirements for the geometry of the roof of the goaf in the design. Due to blasting and other reasons, local over-and under-excavations exist, resulting in the irregular shape of the roof. In addition, the filling method is inappropriate, forming a relatively natural slope angle at the blanking point, and after the blanking opening is blocked at the top of the filling body, some goafs cannot achieve the predetermined goal.

To date, for high-grade ore mining, stope structure tends to be narrow and long in order to improve mining recovery. Narrow long stope refers to the stope with the ratio of stope length to stope width (ratio of length to width) greater than 5~10 by the layered filling method or subsequent filling method. The southeast orebody (760ML-3MU-3 #) in the Chambishi Copper Mine, Zambia, China is segmented and subsequently filled with a length of 123 m, a width of 9 m, a height of 7–9 m, and a aspect ratio of about 14 [62]. It is a great challenge for filling slurry to roof connection once and effectively in such a long mining field with a large aspect ratio.

### 3.2.3. Objective Factors

(1)  Human factors

In traditional engineering design and construction, they are often conducted through experience. Due to different engineering conditions, there will be a lot of blind performances in the construction process, resulting in simple filling facilities and unreasonable hanging of backfilling pipelines, resulting in a poor backfilling effect and multiple backfilling operations, making it difficult to ensure the stability of the backfilling massif, prolonging the production cycle and increasing the backfilling cost. In addition, the experience level of operators, the understanding of operation time, and the adaptability to backfilling technology also affect the roof-contacted backfilling rate.

(2)  Limited auxiliary leveling measures

The backfilling process of goaf is similar to the common concrete-pouring operation, which also requires the pouring slurry to fill the predetermined space. However, in the process of backfilling, the underground stope is closed and the auxiliary leveling measures are limited. Under the existing technical conditions, the roof connection backfilling technology is similar to being performed in a "black box", which has a certain impact on the effect of the roof-contacted backfilling. It is difficult to perform manual auxiliary leveling, such as concrete pouring, and it can only rely on self-leveling. The operators cannot perceive the actual situation of the backfilling slurry, so they can only wait until the solidifying reaches the specified age, check the stope shape, adjust the position of the filling pipeline, and perform subsequent filling operations [63,64].

(3)  "One-time" operation

The filling facilities are arranged in advance, and the backfilling pipeline is hung at the designated position of the roof of the underground stope according to the design

requirements. After constructing the backfilling retaining wall, the operators exit the area to be filled and conduct the backfilling operation. Therefore, once the relevant backfilling equipment is determined, it cannot be moved and regulated at will during the backfilling operation. Therefore, the backfilling operation is called a "one-time" operation, and the layout parameters of backfilling equipment directly affect the roof-contacted effect of the filling body.

In order to improve the roof-contacted backfilling rate, this section sorts out and analyzes the influencing factors of the characteristics of filling slurry, stope characteristics, and related objective conditions. The structural diagram of the influencing factors of the roof-contacted backfilling rate is shown in Figure 10.

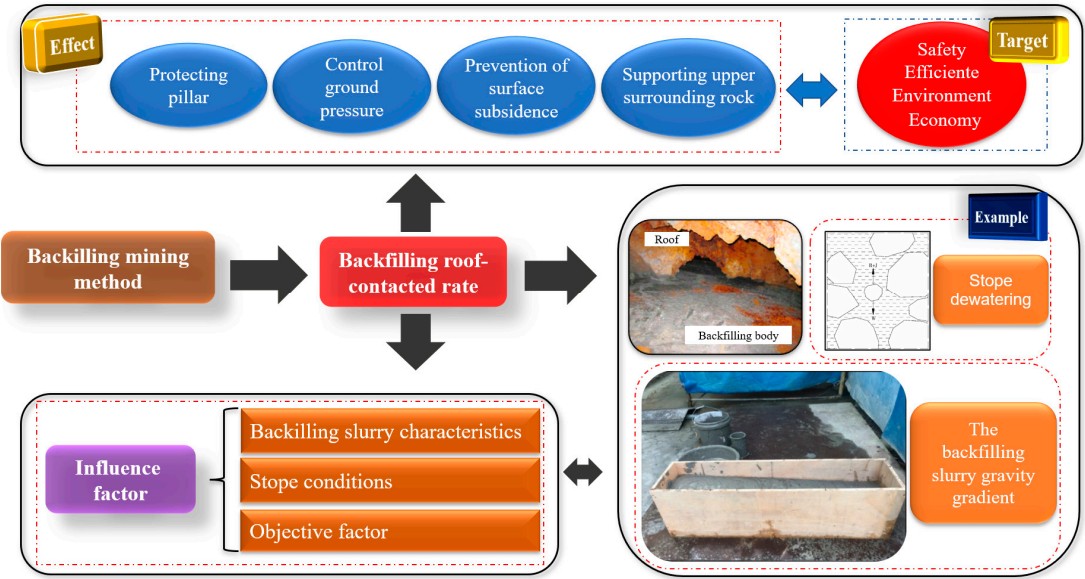

**Figure 10.** Structural diagram of influencing factors of roof-contacted rate.

### 3.3. Improvement Measures

### 3.3.1. Expansion and Non-Shrinkage Material

Compared with the traditional passive roof connection, the backfilling technology of expansion and non-shrinkage material is an effective method. When the backfilling slurry enters the goaf, the volume of the backfilling massif expands in a short time by using its expansion performance, so as to achieve the effect of backfilling massif roof connection. Expansion filling materials can be divided into two types according to different expansion sources. One is the gas-phase expansion caused by an external expansion agent (foaming agent) in a physical way and the generation of bubbles in the slurry through a chemical reaction; the other is solid expansion. Chemical foaming refers to the chemical reaction between a foaming agent and corresponding substances in the backfilling slurry material to release gas. With the condensation of the backfilling slurry material, the gas forms bubbles in the backfilling slurry material, which promotes the expansion of the backfilling slurry volume

(1) Expansive filling material

Inspired by expansive concrete, filling workers at home and abroad introduced the expansion technology of cement-based materials into mine backfilling [65–67]. While selecting appropriate filling materials and the ratio, they modified the filling slurry to cause it to have a certain expansion performance [68,69]. It is a rational idea to adopt the expansion and non-shrinkage material filling technology in the backfilling process. Lan Wentao et al. [70] used HPG (hemihydrate phosphogypsum) as the main raw material, and used its gelling activity to prepare a new type of multiphase, condensable, water-swelling material. The material is mainly composed of four materials: HPG (hemihydrate

phosphogypsum), SAP (amorphous coagulant), GPA (gas phase introduction agent), and HA (hydrophobic agent). The formed filling body is solid, liquid, and gas. The outstanding three-phase features are solidification under the condition of a non-solid volume ratio of 87.6%; high early strength; and, with a certain expansion performance, it can realize the "active roof connection" of the backfilling massif.

Bentonite is mainly composed of montmorillonite clay minerals and belongs to a natural pozzolanic material. It can be divided into three types: nano bentonite, calcium bentonite, and organic bentonite [71]. Among them, sodium bentonite has the characteristics of high dispersion, high water absorption, and multiple large expansions (20~30 times). Using this characteristic, adding it to the backfilling material can expand the backfilling massif. In 1995, Professor Siriwardane et al. [72]. discussed the problem of adding fly ash filling material into bentonite to avoid overburden, collapse, and land subsidence. Through indoor experiments, numerical simulation, and large-scale field practice, the results show that, after adding bentonite, the fly ash filling material has good adhesion and rheological properties, so that the slurry can be filled into the goaf smoothly, and the backfilling slurry has certain expansion properties. Satter Barat et al. [73] mixed bentonite with tailings to study its strength performance. The test proved that bentonite could be used as backfilling material. Bentonite not only has a good expansion performance, but also has significant adsorption on heavy metal ions, which has great environmental benefits.

(2)　Foaming expansion filling material

Based on the research results in the field of foamed mortar, it has been introduced into the filling field and achieved a good backfilling and roof-connection effect [74,75]. After the foaming agent is mixed with other filling materials, the filling slurry produces a strong alkali-solution environment, and the foaming agent produces tiny bubbles in the strong alkali environment, as shown in Figure 11. According to the way that filling materials produce bubbles, they can be divided into chemical foaming and physical foaming. Physical foaming is made by mechanical agitation or foaming agent, which has a certain tension of foam, and then the foam is blown into the slurry. With the condensation of the slurry, an expansion material with uniform porosity is formed. Chemical foaming refers to the chemical reaction between the foaming agent and the corresponding substances in the filling slurry material to release gas. With the condensation of the slurry material, the gas solidifies in the slurry material to form bubbles, which expand the volume of the backfilling slurry.

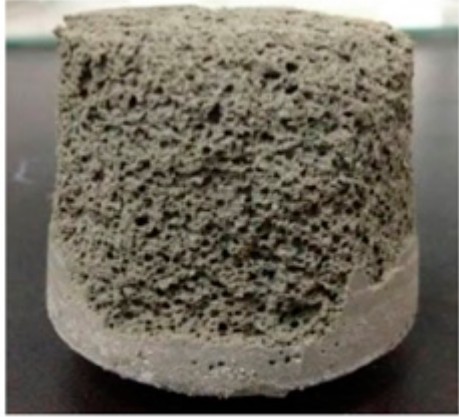

**Figure 11.** The pores in backfilling body.

Taking hemihydrate phosphogypsum as a raw material, Rong K et al. [76] studied the performance-change law of hemihydrate phosphogypsum expansive material under the combined action of multiple factors, and conducted random tests. The results show that increasing the parameters of gas-phase air entraining agent can increase the expansion rate of backfilling slurry, which is more conducive to the roof-contacted backfilling massif,

but reduces the strength and durability of the filling body. Adnan Colak et al. [77] used sodium lauryl sulfate and nonylphenol ethoxylated foaming agent to produce bubbles in gypsum. In order to promote foam and bubble formation, retarder citric acid and tackifier carboxymethyl cellulose were used. The results show that foams or bubbles are only intermediate products of expansive mortar materials, and the ultimate goal is that the volume of mortar after filling will result in volume expansion due to the bubbles formed inside the mortar. When a variety of expansion agents, such as foaming agents, are added to fill the filling slurry, the phenomenon of first shrinkage and then expansion can be achieved in the process of solidification and condensation. Compared with the final volume without an expansion agent, the filling slurry expands, as shown in Figure 12.

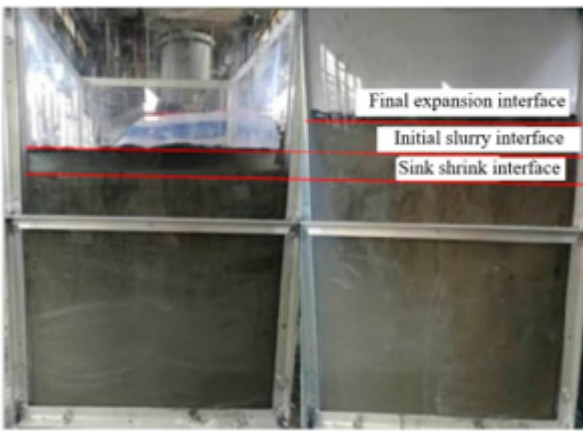

**Figure 12.** Effect of backfilling body with expansion agent.

In mine backfilling, due to the addition of expansive agent, the internal structure of the backfilling massif is relatively weak, and the increase in volume is due to the decrease in volume density, namely, the increase in porosity. In the backfilling massif expansion technology, the expansion of the backfilling massif is often accompanied by the deterioration of the strength of the backfilling body, and the high-dose expansive agent causes fatal damage to the stability of the backfilling massif. In the stope with a large aspect ratio, the slurry flow forms a certain sedimentary slope, and the later expansion leads to partial roof connection. Therefore, only relying on expansive filling cannot completely deal with the problem of filling body roofing.

### 3.3.2. Forced Roof-Contacted Measures

When the above methods are not enough to achieve the predetermined backfilling target, it is a favorable way to apply the improvement measures in Table 2, but each of these improvement measures is restricted by goaf condition and can only solve specific situations.

### 3.3.3. Strengthen-Management Level

Due to the limitation of goaf conditions, the uneven concave or convex shape of the stope roof (local over-excavation and under-excavation), the inconsistent filling sequence, the mixing of pipe washing water, and the randomness of manual operations affect the roof-contacted rate of the filling body. The ratio of filling slurry can be optimized and its roof connection performance can be increased by optimizing the filling and discharge process, means of standardization of operation parameters, and other processes. Pay more attention to filling quality management and hire professionals engaged in filling management. After finding the problems, take the initiative to perform some rectification and disposal measures. Moreover, achieve on-site supervision and tracking, and perform reasonable disposal and accurate analysis, and the evaluation of on-site problems, as shown in Table 3.

**Table 2.** Partial improvement measures.

| Forced Roof-Contacted Measures | Characteristic |
|---|---|
| Manual roof connection | High labor intensity, low work efficiency, and poor working conditions. |
| Mechanical roof connection | It is used in metal mines, such as the segmented filling method and route filling method. It is mainly used in the field of cemented slurry of high-concentration coarse aggregate and non-cemented filling of waste rock. |
| Forced-caving roof connection | It is a common method for slightly and gently inclined ore bodies. |
| Natural-caving roof connection | The physical requirements for the high performance of ore and rock are guaranteed by a high management level.<br>Applied to its own low-strength characteristics and ore bodies with joint fissure distribution. |
| Slurry-pressure roof connection | It is mainly used for the up- and down-filling mining methods.<br>It is divided into residual pressure roof-connection of the filling system and pressure pump injection roof-connection. |
| Slurry self-flowing roof connection | Use the slurry to level under its own gravity or use the height difference to extrude the slurry to connect the roof. |

**Table 3.** Ways to improve management.

| Operating Time | Measures |
|---|---|
| Before backfilling operation | Creating good conditions for stope-filling top pick.<br>Select the appropriate slurry concentration and packing materials.<br>The use of intumescent material additives. |
| During backfilling operation | Eliminating the influence of water.<br>Noting the empty top pressure.<br>Reducing worker error and strictly quality-controlling projects. |
| After backfilling operation | Leakage of slurry is prohibited.<br>Prevent the influence of water on slurry.<br>Improving roof monitoring. |

In order to improve the backfilling roof-contacted rate, this section summarizes the improvement measures in three aspects: using expansive materials, forced roof-connection technology, and strengthening management. The structure chart of influencing factors of roof-connection improvement measures is shown in Figure 13.

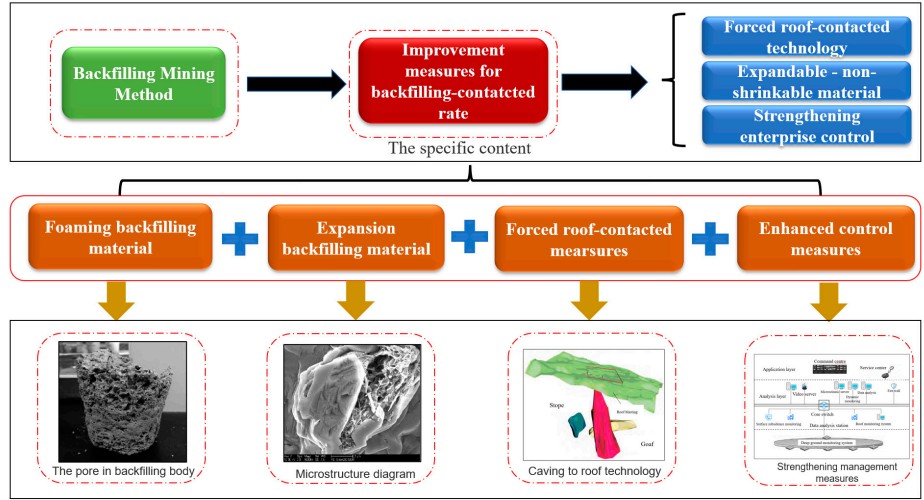

**Figure 13.** Influence structure of roof-contacted improvement measures.

## 4. Conclusions

(1) Backfilling technology is an important technical method for creating a win–win situation of "environmentally friendly" and "safe and efficient" mines. Roof-contacting backfilling is the key factor of the backfill mining method, which is directly related to the support capacity of upper surrounding rock and guarantees the safety of stope. With the increase in mining depth and the deterioration of mining conditions, more attention must be paid to the roof-contacted backfilling rate in the future.

(2) The roof connection of the backfilling massif is an important part of backfilling work. In this paper, the method of calculating similar roof-contacted backfilling rates was introduced. For conventional high-concentration cementitious backfilling, the main influencing factors, improvement measures, and auxiliary measures of the roof-contacted backfilling rate were summarized in detail.

(3) It is still a challenge for the backfilling massif to connect to the roof efficiently in a mining field with a large aspect ratio. During the flow process of filling slurry in the underground stope, the yield surface position dynamically changes, and there is no directly test method to detect the yield surface position. In future research, in the process of backfilling slurry flow in underground stopes, relevant research on the position of the slurry yield surface should be strengthened to make up for the deficiency of theoretical models in parameter corrections.

(4) The roof connection of the backfilling massif is a systematic project. In the design of the roof-connection scheme, stope design, mining process requirements, slurry performance, and roof auxiliary technology should be compared and selected.

**Author Contributions:** Conceptualization, F.C. and H.J.; methodology, J.W.; formal analysis, X.Z.; investigation, H.J. and X.Z.; writing—original draft preparation, J.L.; writing—review and editing, H.J., F.C. and J.Y. All authors have read and agreed to the published version of the manuscript.

**Funding:** This research was funded by the National Natural Science Foundation of China (No. 51834001).

**Data Availability Statement:** Data is contained within the article.

**Conflicts of Interest:** The authors declare no conflict of interest.

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
