# Peer review of "Review on the Art of Roof Contacting in Cemented Waste Backfill Technology in a Metal Mine"

_minerals, doi:10.3390/min12060721_

Round 1

Reviewer 1 Report

The presented article refers to the coverage of important problematic issues of the goaf backfilling processes, which today allows for "green mining". A particularly problematic place in filling the worked-out space is the contact of the backfilling mass with the roof, which can affect the slump of the backfilling mass. The authors present their opinion on the factors influencing these processes, as well as ways to solve the problem of backfilling under the roof.

It should be noted that the article is of a review type, not a research one.

The material of the article certainly corresponds to the theme of the special issue of the journal.

However, after reading the article in detail, I had several discussion questions and recommendations for improving the article.

  1. There is no statement of the research problem. The introduction does not cover the problem of the construction of the backfilling massif and its full contact with the roof of the rock massif. The introduction deals in general with the importance of backfilling for the safe development of mineral deposits. This is noted in section 2. I recommend that this be attributed to the introduction or section 2 should be called "Backfilling problems ...".

  1. From the text of the article it is completely unclear for which specific mines the problem of backfilling under the roof is considered? Under which mining methods do the problems under consideration arise? (sublevel stoping and other) It is recommended to clarify this.
  2. Figure 3 is of poor quality, it is recommended to rebuild it.

  1. It is not clear what are the parameters of the goaf to be backfilled? For what sizes of goaf is the problem and proposed solutions. It may be necessary to consider an example of a problem in a particular mine environment.

  1. The conclusions are well known regarding backfilling of the mined-out area. It is necessary to reflect the results obtained on the basis of the analytical work carried out, your own conclusions on the problem under consideration. It is necessary to work on the conclusions and present more specific ones, obtained as a result of analysis and deductions.

Reviewer 2 Report

The manuscript "Research Progress on high efficiency roof joint filling technology of mine solid waste" by Fengbin Chen, Jiguang Liu, Xiaowei Zhang, Jinxing Wang, Huazhe Jiao, Jianxin Yu was submitted for review.

The authors considered a very topical subject: increasing the safety of mining operations and reducing the burden on the environment through the introduction of closed production through the use of man-made waste. Presented study is interesting and may constitute a valuable contribution to the area of knowledge. It may have practical applications in the mining industry.

At the same time, I would like to note that the manuscript has a number of minor shortcomings, the elimination of which will improve the scientific quality and increase the perception of presented information.

1.) From my point of view, the literature review is rather poorly presented in the text of previous studies

1.1.) It is necessary to more clearly outline the problems raised by the authors (the moment when they began to raise this problem in science and this topic, and if they didn’t, then why?)

1.2.) It is necessary to summarize the main issues that other scientists have previously disclosed (or the main thing they omitted and why?), maybe they did not directly affect this problem (and why?), but it can be traced in works in related topics, and then lead to the tasks of the article itself are to close the blank spot.

1.3.) In lines 41 - 44; 47 - 56; 143 - 146; 155 - 165 the authors make a number of statements that are not confirmed by the corresponding references.

2.) From my point of view, it is necessary to briefly summarize the analysis of previous work and outline the goals and objectives of this work. For example: "The development of a technology for laying technogenic waste under the roof of a working, which increases the safety of mining operations, which will reduce the burden on the environment, seems to be a very urgent issue. In this regard, the purpose of this work is ...... To achieve the goal, it is necessary to solve the following tasks 1)...2)..."

3.) Figure 5 shows the distribution of the suspension in the tailings basin depending on the particle size. It is not clear to me on what basis the authors suggested that this is how the distribution occurs. It is necessary to indicate which laws or sources the authors used. If this image is borrowed, is there permission from the copyright holders to demonstrate it.

4.) In line 254, the authors talk about reaching a "specified age". What age are we talking about?

5.) The authors' claim that the addition of expanders does not affect the strength characteristics of the composite after it has hardened (line 293) is highly controversial. This statement does not follow and is not proved in the paper [ref. 45]. The increase in volume occurs due to a decrease in bulk density, that is, due to an increase in porosity. As a rule, a less dense material (with a large number of pores) has reduced strength characteristics. Also, the authors did not study the strength characteristics of the samples.

6.) In line 348-350, the authors talk about the need to optimize the filling slurry. What then did the authors do in the course of this study? As I understand it, they set the task of laying the chamber under the roof of the excavation. How can this be done if we do not optimize the volume of the expansion agent?

7.) Statements in line 350-360 are too abstract for scientific work. What combination can achieve the maximum filling of the roof opening? What improvement measures need to be taken? How to improve the level of management?

8.) There is no section on methods and materials. It is not clear to me what methods the authors used and what materials were taken for the experiments. In order for other scientists to repeat these experiments, it is necessary to indicate the material and fractional composition of the waste used.

9.) From my point of view, the use of tailings as an inert aggregate without reprocessing is a throwback. From my point of view, it is necessary to point to innovative technologies that reduce the amount of a valuable component in industrial waste to minimum. I’ve seen many contributions published recently in Minerals, concerning mineral extraction from tailings. Please check in MDPI search engine and, maybe, make some reservations concerning this aspect.

10) There are a lot of harmful components in the enrichment tailings. The design of tailings prevents the rapid penetration of harmful components into the environment. However, when these harmful components enter the bowels, they become quite mobile. Mobility is facilitated by a system of cracks and groundwater. The authors did not indicate what measures are possible to reduce the mobility of harmful components.

11) 11.1. there is no information on the method of preparation of the samples, namely: how the mixing took place (order, what was poured after), the equipment on which the mixing took place and the speed of mixing. In addition, there are components in the composition, the content of which is unequal, in connection with this I have a question: how was their uniform distribution achieved in the entire mass of the composite

11.2. there is no information under what conditions they were stored. To what extent do these storage conditions correspond to underground ones? There is no reference to the method of preparation and testing or similar works, where there is a description of this method, taking into account international experience.

     11.3. the authors did not indicate the physical and mechanical properties of the resulting material. Are the strength characteristics sufficient to be used to maintain the treatment space.

     11.4. How was convergence achieved?

 Otherwise, other researchers do not have the opportunity to repeat this experiment.

12.) Conclusions do not comply strictly to the research

12.1.) The statements of the authors "the technology of backfilling mine workings reduces the open areas of an empty area underground, controls the movement of the rock layer" is not directly supported by research. The authors did not conduct studies confirming this statement.

12.2.) Conclusion (2) is not a conclusion, but more related to the problem that has currently arisen with the transition of mining to greater depths

12.3.) Conclusion (3) also does not follow from the present study

12.4.) I would like to think that the conclusion (4) can be attributed to the formulation of the problem for further research. But it's not!

The conclusions need to be cautiously rewritten. I’d suggest to soften the statements and make some reservations. Authors work is of pioneer nature, so please provide some prospects for future research and development.

I have an impression that too strong conclusions in this study are associated with the lack of correctly set goals and objectives for its achievement (remark 2). From my point of view, in connection with this, the authors themselves brought confusion into their work, but it can be easily improved.

This work touches on a fairly relevant topic and can be published after making relevant changes in accordance with the indicated comments. Please feel free to discuss my opinion. Best value in science is always a result of discussion.

Round 2

Reviewer 1 Report

The authors did a good job of correcting the comments. The recommendations allowed to improve the quality of the article. The article has a presentable appearance and is more understandable for the reader.

Author Response

Dear Reviewer 

Once again, thank you very much for your comments and suggestions.Your comments are of great help to our future research.

Best regards,

Yours sincerely,

Jinxing Wang

Reviewer 2 Report

Dear Authors

Thank you very much for the time that you devoted to improve your manuscript. I find it now suitable for publication in Minerals.

Sincerely

Author Response

(The authors gave the same response as above.)
